# Derivation and validation of a predictive model for chronic stress in patients with cardiovascular disease

Ali O. Malik[1,2], Philip G. Jones[1,2], Carlos Mena-Hurtado[3], Matthew M. Burg[3], Mehdi H. Shishehbor[4], Vittal Hejjaji[1,2], Andy Tran[1,2], John A. Spertus[1,2], Kim G. Smolderen[3]*

1 Saint Luke's' Mid America Heart Institute, Kansas City, MO, United States of America, 2 University of Missouri Kansas City, Kansas City, MO, United States of America, 3 Yale School of Medicine, New Haven, CO, United States of America, 4 Case Western Reserve University, Cleveland, OH, United States of America

* kim.smolderen@yale.edu

## Abstract

### Background

Chronic stress in patients with cardiovascular disease (CVD), including peripheral artery disease (PAD), is independently associated worse outcomes. A model that can reliably identify factors associated with risk of chronic stress in patients with CVD is needed.

### Methods

In a prospective myocardial infarction (MI) registry (TRIUMPH), we constructed a logistic regression model using 27 patient demographic, socioeconomic, and clinical factors, adjusting for site, to identify predictors of chronic stress over 1 year. Stress at baseline and at 1-, 6- and 12-month follow-up was measured using the 4-item Perceived Stress Scale (PSS-4) [range 0–16, scores ≥6 depicting high stress]. Chronic stress was defined as at least 2 follow-up PSS-4 scores ≥6. We identified and validated this final model in another prospective registry of patients with symptomatic PAD, the PORTRAIT study.

### Results

Our derivation cohort consisted of 4,340 patients with MI (mean age 59.1 ± 12.3 years, 33% females, 30% non-white), of whom 30% had chronic stress at follow-up. Of the 27 factors examined, female sex, current smoking, socioeconomic status, and economic burden due to medical care were positively associated with chronic stress, and ENRICHD Social Support Instrument (ESSI) score and age were inversely related to chronic stress. In the validation cohort of 797 PAD patients (mean age 68.6±9.7 years, 42% females, 28% non-white, 18% chronic stress) the c-statistic for the model was 0.77 and calibration was excellent.

### Conclusions

We can reliably identify factors that are independently associated with risk of chronic stress in patients with CVD. As chronic stress is associated with worse outcomes in this population,

on health outcomes. For the PORTRAIT study data requests can be sent to Yale University Institute Review Board, at hrpp@yale.edu. For the TRIUMPH study data requests could be sent to the steering committee for the TRIUMPH registry at dbuchanan@saint-lukes.org.

**Funding:** Drs. Malik, Hejjaji and Tran are supported by the National Heart, Lung, And Blood Institute of the National Institutes of Health under Award Number T32HL110837. Dr. Mena is a consultant for COOK, Medtronic, Cardinal Health, Optum Labs. Dr. Shishehbor is on the advisory board of Medtronic, Abbott Vascular, Terumo, Boston Scientific, and Philips. Dr. Burg is supported in part by grants from the National Heart, Lung and Blood Institute under award numbers R01HL125587, R01HL126770, and R01HL152548. Dr. Spertus owns the copyright to the PAQ and SAQ and, unrelated to this work, is the principal investigator of an analytic contract from the American College of Cardiology Foundation to provide analytic services for the National Cardiovascular Data Registries, provides consultative services to Novartis, Bayer, AstraZeneca, Janssen, Merck, United Healthcare and Amgen; owns the copyright to the KCCQ; serves on the Board of Directors of Blue Cross Blue Shield of Kansas City; and has an equity interest in Health Outcomes Sciences. Dr. Smolderen reports support through an unrestricted research grant from Terumo and she is a consultant for Optum Labs. The funders had no role in study design, data collection and analysis, decision to publish, or preparation of the manuscript.

**Competing interests:** The authors have declared that no competing interests exist.

our work identifies potential targets for interventions to as well as the patients that could benefit from these.

## Background

High stress levels have been associated with development of cardiovascular disease (CVD) [1], and in patients with CVD, with adverse outcomes, including death [2,3], recurrent events [4,5] and poorer quality of life [2]. The latest American College of Cardiology and American Heart Association guidelines on prevention of CVD recommend addressing psychosocial stressors as a preventive measure to decrease cardiovascular risk [6]. Randomized controlled trials testing interventions to mitigate the impact of stress, including cognitive behavioral therapy [7], transcendental meditation [8], and group psychotherapy sessions [9] have demonstrated a decreased risk of death and recurrent events in patients with CVD. However, the results of these trials have not been widely adopted in clinical practice. Strategies to prevent worse outcomes in CVD could be enhanced by integration of interventions specifically targeting chronic stress into the broader context of cardiovascular care. A prerequisite to his goal is to understand the totality of patient and societal factors contributing to the risk of chronic stress in patients with CVD.

To our knowledge, no risk model has been validated to predict chronic stress in patients with CVD. Such a model is needed to understand the impact of factors that play a role in the development of chronic stress in patients with CVD and to identify patients who may benefit from future programs that address these underlying factors directly in conjunction with cardiovascular rehabilitation programs with integrated care pathways to help manage chronic stress. We aimed to develop such a model in a cohort of patients who survived acute myocardial infarction (AMI) and to validate the model in a cohort of patients with peripheral artery disease (PAD).

## Methods

The investigators are willing to work with others, who are interested in validating or extending our analyses. For the PORTRAIT study data requests can be sent to Yale University Institute Review Board, at hrpp@yale.edu. For the TRIUMPH study data requests could be sent to the steering committee for the TRIUMPH registry at dbuchanan@saint-lukes.org.

### Study population

Data from a prospective registry of patients presenting with AMI, the Translational Research Investigating Underlying disparities in acute Myocardial infarction Patient's Health Status (TRIUMPH) study was used for model derivation [10]. For external validation of our model, we used data from a prospective registry of patients presenting with worsening Peripheral Artery Disease (PAD), the Patient-centered Outcomes Related to Treatment Practices in Peripheral Arterial Disease: Investigating Trajectories (PORTRAIT) registry [11]. Although both TRIUMPH and PORTAIT included patients who had a different presentation of CVD, both included a patient population who had worsening of their CVD and examined trajectories of stress using the same instrument to quantify stress, social support, as well as similar patient demographic, psychosocial, clinical and socioeconomic measures.

**Registry designs.** The design of both the TRIUMPH and PORTRAIT study have been published elsewhere [10,11]. The TRIUMPH study enrolled patients (n = 4,340) presenting

with MI who were alive at hospital discharge. Enrollment was done form April 11, 2005 to December 31, 2008 across 24-US hospitals. Patients who were enrolled had biomarker evidence of myocardial necrosis and additional clinical evidence supporting the diagnosis of AMI, including prolonged ischemic signs/symptoms or electrocardiographic criteria of ST segment changes. Baseline data were obtained through chart abstraction and structured interviews by trained research coordinators. Data on health status and psychosocial stress were obtained at baseline, 1-, 6- and 12-month follow-up using a standardized interview conducted by trained study personnel.

The PORTRAIT registry enrolled patients who presented with worsening symptoms of PAD to sub-specialty clinics in the US (n = 797), Australia (n = 95) and Netherlands (n = 383) from June 2011 to December 2015. For this study, only the patients from the US were included. Enrolled patients had an ankle brachial index (ABI) $\leq$ 0.90 or a significant drop in post-exercise ankle pressure ($\geq$ 20mm of Hg). Patient demographics, health status, psychosocial characteristics, socioeconomic variables, and cardiovascular lifestyle factors were obtained through interviews at the initial visit. Patient symptoms, medical history, comorbidities, and PAD diagnostic information were abstracted from medical records. Serial information about health status and patient psychosocial profile was collected at baseline, 3-, 6- and 12-month follow-up through centralized follow-up. For both TRIUMPH and PORTRAIT studies, all study participants provided written or telephonic informed consent and the study protocol was approved by Institution Review Boards of Saint Luke's Hospital and all participating sites.

**Assessment of stress and definition of chronic stress.** In both TRIUMPH and PRE-MIER, level of perceived stress was assessed at enrollment and follow-up with the 4-item perceived stress scale (PSS-4). The PSS-4 is a reliable and valid measure (Cronbach's Alpha 0.67–0.79) of an individual's self-evaluation of control and confidence in handling the stressful situations they have experienced over the past month [12]. In our study, the Cronbach's Alpha was 0.79 for the TRIUMPH cohort and 0.68 for the PORTRAIT cohort. Scores on the PSS-4 range from 0–16, with higher scores indicating higher stress and lower ability to cope with that stress [12]. The PSS-4 is a non-diagnostic instrument and there are no established thresholds, although in patients with cardiovascular disease, a score of $\geq$6 has been associated with adverse outcomes in patients after MI [2]. Hence, in keeping with prior research we used a score of $\geq$6 as the threshold to describe high levels of perceived stress. PSS-4 was collected at baseline and at each follow-up assessment. We wanted to quantify a patient's exposure to chronic stress, during the 12-months of follow-up. Therefore, to provide more stable categorization of stress levels at follow-up we defined chronic stress as 2 or more follow-up PSS-4 assessments of $\geq$6, after the initial baseline assessment. As the initial event (AMI or worsening PAD symptoms) could contribute to the patient's stress in the first few days, baseline PSS-4 assessments were not included in the definition.

**Assessment of socioeconomic status.** General socioeconomic status (SES) was assessed using the question, "how much money do you have left over at the end of the month?" with possible responses being "enough", "just enough" and "not enough". Economic burden due to medical care was assessed using the question, "What is the economic burden of your medical costs" with possible responses being "severe burden", "moderate burden", "somewhat of a burden", "a little burden" and "no burden at all".

**Social support and disease-specific health status.** Social support was quantified using the ENRICHD Social Support Instrument (ESSI), which has been derived from the Medical Outcomes Survey and prior work examining the influences of social support [13]. The ESSI is a 7-item measure and assesses four attributes of social support: emotional, instrumental, informational and appraisal [13,14]. The ESSI was found to be a valid and reliable measure of social support in patients to screen for patients enrolled in a depression intervention trial [15].

**Statistical analysis.** Patient demographic, socioeconomic, clinical factors and PSS-4 scores at baseline were described separately in our derivation and validation cohorts. To identify predictors of chronic stress over 1 year, we constructed a multivariable hierarchical logistic regression on all patient factors (listed in Table 1), adjusting for study site as a random effect. No appreciable multicollinearity was found among the predictors (all variance inflation factors < 2.0; design matrix condition index 17.6). Nonlinear effects for continuous variables were examined using restricted cubic splines; however, no significant nonlinearity was detected (p = 0.73), so effects were refit linearly for parsimony and ease of interpretation. There was moderate site-level variability in the outcome of chronic stress over 1-year (Median Odds Ratio 1.21), and to account for this variability we added site as a random effect in the model. Model performance was assessed using the c-statistic to determine discrimination and by plotting deciles of predicted risk against the observed event rate and comparing the regression line with the line of unity (intercept = 0 and slope = 1). Finally, to understand the prevalence of independent predictors of chronic stress in patients stratified by age, we compared the prevalence of predictors in patients <55 and ≥ 55 years of age.

**Missing data.** Of the 4,340 patients in TRIUMPH, 1,682 had complete PSS-4 scores at three follow-up assessments, 1,105 at only two assessments, 786 at only one assessment, and 767 had no follow-up scores. Scores were missing due to skipped items (2.8%), refusals (4.1%), illness (2.4%), lost to follow-up (24.8%) or death (4.0%). We used multiple imputation by chained equations (MICE) with predictive mean matching to impute missing PSS-4 scores (as well as missing values of candidate predictor variables) [16]. The imputation model included all available PSS-4 questionnaire items at all time points, site as well as the 27 predictors of interest that we identified a priori based on previous literature and clinical judgement. A total of 20 randomly imputed data sets were generated, and the outcome of chronic stress was defined on each data set using observed and imputed scores. All analyses were performed by analyzing each of the 20 data sets separately and then pooling the results, to account for bias and uncertainty due to missingness.

## Results

### Patient populations

In the derivative cohort (TRIUMPH study), the mean age of the study population was 59.1 ±12.3 years, 33.2% were females and 30.2% were non-white. Overall, 30% of the patients had chronic stress at follow-up. Table 1 describes baseline patient characteristics for the candidate variables in the derivation cohort and compares prevalence of chronic stress among patients stratified by their baseline characteristics. Patients in age groups of 19–54 years had the highest prevalence of chronic stress. Moreover, prevalence of chronic stress was higher in non-whites, in patients who reported not having enough finances at month's end, and in patients who perceived healthcare costs to be a severe economic burden.

In the validation cohort (PORTRAIT study), the mean age was 68.6 ± 9.7 years, 41.9% of the patients were females and 27.6% were non-white. Overall 18% of the patients had chronic stress at follow-up. Table 2 describes the baseline patient characteristics and prevalence of chronic stress among patients stratified by each baseline factor. Patients aged 42–54 years, females, non-whites, and patients who reported end of month financial distress and severe burden of healthcare costs had a higher prevalence of chronic stress.

### Predictive model

Fig 1 describes the baseline patient factors evaluated in the model. Of all the baseline patient factors examined (Fig 1), 6 were found to be independently associated with outcome of

**Table 1. Baseline patient characteristics and prevalence of chronic stress in patients from the TRIUMPH study.**

| | Total n (%) | Prevalence of Chronic Stress | p-value |
|---|---|---|---|
| *Demographics* | | | |
| Age (years) | | | <0.001 |
| 19.0 to <55 | 1633 (37.6%) | 40.4% | |
| 55 to <65 | 1374 (31.7%) | 31.8% | |
| 65 to <75 | 782 (18.0%) | 17.8% | |
| 75 to 98.0 | 551 (12.7%) | 19.5% | |
| Male | 2898 (66.8%) | 28.1% | <0.001 |
| Female | 1442 (33.2%) | 36.7% | |
| Non-white | 1305 (30.2%) | 38.0% | <0.001 |
| White | 3022 (69.8%) | 27.9% | |
| *Socioeconomic Factors* | | | |
| Married | 2318 (53.5%) | 24.6% | <0.001 |
| Not Married | 2014 (46.5%) | 38.2% | |
| *Finances at the End of the Month* | | | <0.001 |
| Some money left over | 1777 (41.7%) | 15.7% | |
| Just enough to make ends meet | 1592 (37.4%) | 34.7% | |
| Not enough to make ends meet | 889 (20.9%) | 54.6% | |
| Not Working | 2200 (51.2%) | 34.8% | <0.001 |
| Working | 2100 (48.8%) | 26.9% | |
| *Education* | | | <0.001 |
| Less than high school | 895 (20.7%) | 38.9% | |
| High school | 2542 (58.9%) | 31.3% | |
| College degree | 878 (20.3%) | 21.8% | |
| Has avoided care due to cost | 1088 (25.6%) | 51.2% | <0.001 |
| Has not avoided care due to cost | 3165 (74.4%) | 24.0% | |
| *Medical Costs Economic Burden* | | | <0.001 |
| Severe burden | 447 (10.5%) | 55.1% | |
| Moderate burden | 409 (9.6%) | 46.4% | |
| Somewhat of a burden | 507 (11.9%) | 38.9% | |
| A little burden | 440 (10.3%) | 30.3% | |
| No burden at all | 2462 (57.7%) | 22.4% | |
| Lives alone | 1061 (24.6%) | 34.5% | 0.023 |
| Does not live alone | 3247 (75.4%) | 29.8% | |
| *ESSI score* | | | <0.001 |
| 5 to <20 | 886 (21.1%) | 52.0% | |
| 20 to <25 | 1328 (31.6%) | 28.9% | |
| 25 to 25 | 1985 (47.3%) | 22.8% | |
| *Comorbid Medical Conditions* | | | |
| Body Mass Index (kg/m$^2$) | | | 0.001 |
| 13.5 to <25 | 959 (23.3%) | 29.6% | |
| 25 to <30 | 1467 (35.7%) | 27.2% | |
| 30 to <35 | 974 (23.7%) | 32.1% | |
| 35 to 70.7 | 709 (17.3%) | 39.0% | |
| *Smoking Status* | | | <0.001 |
| Current | 1689 (39.2%) | 41.3% | |
| Former | 1403 (32.6%) | 22.8% | |
| Never | 1215 (28.2%) | 25.9% | |
| *Hypertension* | | | 0.008 |
| Yes | 2893 (66.7%) | 32.4% | |
| No | 1447 (33.3%) | 28.1% | |
| *Diabetes* | | | 0.006 |
| Yes | 1336 (30.8%) | 34.9% | |
| No | 3004 (69.2%) | 29.2% | |
| *Dyslipidemia* | | | 0.37 |
| Yes | 2128 (49.0%) | 30.2% | |
| No | 2212 (51.0%) | 31.7% | |

*(Continued)*

**Table 1.** (Continued)

| | Total n (%) | Prevalence of Chronic Stress | p-value |
|---|---|---|---|
| *Prior Percutaneous Coronary Intervention* | | | 0.18 |
| Yes | 851 (19.6%) | 33.2% | |
| No | 3489 (80.4%) | 30.4% | |
| *Prior Coronary Artery Bypass Graft Surgery* | | | 0.93 |
| Yes | 495 (11.4%) | 31.2% | |
| No | 3845 (88.6%) | 30.9% | |
| *Prior Myocardial Infarction* | | | 0.20 |
| Yes | 912 (21.0%) | 33.0% | |
| No | 3428 (79.0%) | 30.4% | |
| *Prior stroke/transient ischemic attack* | | | 0.40 |
| Yes | 304 (7.0%) | 33.5% | |
| No | 4036 (93.0%) | 30.8% | |
| *Congestive Heart Failure* | | | 0.002 |
| Yes | 372 (8.6%) | 39.9% | |
| No | 3968 (91.4%) | 30.1% | |
| *Atrial Fibrillation* | | | 0.70 |
| Yes | 212 (4.9%) | 32.4% | |
| No | 4128 (95.1%) | 30.9% | |
| *Chronic Kidney Disease* | | | 0.88 |
| Yes | 322 (7.4%) | 31.4% | |
| No | 4018 (92.6%) | 30.9% | |
| *PHQ-8 Depression Score* | | | <0.001 |
| 0.0 to <5 | 2322 (56.8%) | 18.8% | |
| 5 to <10 | 1009 (24.7%) | 35.0% | |
| 10 to <15 | 464 (11.3%) | 55.6% | |
| 15 to 24.0 | 294 (7.2%) | 70.8% | |
| *Baseline stress (PSS-4 $\geq$ 6)* | | | <0.001 |
| Yes | 1622 (38.6%) | 54.2% | |
| No | 2582 (61.4%) | 16.3% | |
| *Hospital Presentation* | | | |
| Non-ST elevation MI | | | 0.003 |
| Yes | 2473 (57.0%) | 33.0% | |
| No | 1867 (43.0%) | 28.2% | |
| In-Hospital Revascularization | | | <0.001 |
| None | 1153 (26.6%) | 37.8% | |
| PCI | 2782 (64.1%) | 28.5% | |
| CABG | 405 (9.3%) | 28.0% | |

ESSI = ENRICHD social support index, PHQ-8 = 8-point Patient Health Questionnaire depression scale, SAQ = Seattle Angina Questionnaire.

chronic stress over 12-month follow-up. S1 Fig details the calibration plot for the 6-item model in the TRIUMPH study. These were age, sex, economic burden related to medical care, general SES, current smoker, and ESSI score. The bootstrapped-validated c-statistic for the final model (including all the 6-items) was 0.75 (S1 Fig). The c-statistic for the final model, applied to the validation cohort (PORTRAIT study) was 0.77 and calibration was excellent (Fig 2). S1 Table gives the intercept and coefficient information for all the covariates in the final regression equation.

## Prevalence of predictors of chronic stress in patients stratified by age

Table 3 shows the prevalence of predictors of chronic stress in our derivative cohort, stratified by age. The socioeconomic predictors of chronic stress were more prevalent in patients younger than 55 years old compared with older patients.

**Table 2. Baseline patient characteristics and prevalence of chronic stress in patients from the PORTRAIT study.**

| | Total n (%) | Prevalence of Chronic Stress | p-value |
|---|---|---|---|
| **Demographics** | | | |
| Age (years) | | | <0.001 |
| 42.0 to <55 | 61 (7.7%) | 46.5% | |
| 55 to <65 | 202 (25.3%) | 27.4% | |
| 65 to <75 | 311 (39.0%) | 11.5% | |
| 75 to 94.8 | 223 (28.0%) | 12.1% | |
| Male | 463 (58.1%) | 14.9% | 0.007 |
| Female | 334 (41.9%) | 23.2% | |
| Non-white | 220 (27.6%) | 25.5% | 0.006 |
| White | 577 (72.4%) | 15.6% | |
| **Socioeconomic Factors** | | | |
| Married | 435 (55.0%) | 14.9% | 0.016 |
| Not Married | 356 (45.0%) | 22.5% | |
| *Finances at the End of the Month* | | | <0.001 |
| Some money left over | 406 (51.4%) | 9.0% | |
| Just enough to make ends meet | 291 (36.8%) | 24.9% | |
| Not enough to make ends meet | 93 (11.8%) | 38.5% | |
| Missing | 7 | | |
| Not Working | 613 (77.5%) | 19.7% | 0.13 |
| Working | 178 (22.5%) | 13.7% | |
| *Education* | | | 0.001 |
| High school | 499 (73.7%) | 32.6% | |
| College degree | 178 (26.3%) | 17.6% | |
| Missing | 120 | 10.8% | |
| Has avoided care due to cost | 130 (16.4%) | 35.2% | <0.001 |
| Has not avoided care due to cost | 661 (83.6%) | 15.0% | |
| *Medical Cost Economic Burden* | | | <0.001 |
| Severe burden | 31 (3.9%) | 46.2% | |
| Moderate burden | 71 (9.0%) | 29.8% | |
| Somewhat of a burden | 86 (10.9%) | 30.8% | |
| A little burden | 95 (12.0%) | 16.4% | |
| No burden at all | 509 (64.3%) | 13.2% | |
| Lives alone | 210 (26.4%) | 19.9% | 0.60 |
| Does not live alone | 586 (73.6%) | 17.8% | |
| *ESSI score* | | | <0.001 |
| 5 to <20 | 139 (17.6%) | 37.2% | |
| 20 to <25 | 221 (27.9%) | 14.9% | |
| 25 to 25 | 432 (54.5%) | 14.0% | |
| **Comorbid Medical Conditions** | | | |
| BMI | | | 0.46 |
| 15.2 to <25 | 186 (24.2%) | 18.7% | |
| 25 to <30 | 270 (35.1%) | 16.1% | |
| 30 to <35 | 192 (24.9%) | 19.5% | |
| 35 to 60.5 | 122 (15.8%) | 21.1% | |
| *Smoke status* | | | 0.003 |
| Never | 104 (13.1%) | 15.1% | |
| Former | 451 (56.7%) | 14.4% | |
| Current | 241 (30.3%) | 27.1% | |
| *Hypertension* | | | 0.99 |
| Yes | 706 (88.6%) | 18.3% | |
| No | 91 (11.4%) | 18.5% | |
| *Diabetes* | | | 0.34 |
| Yes | 305 (38.3%) | 20.6% | |
| No | 492 (61.7%) | 17.0% | |

*(Continued)*

**Table 2.** (Continued)

| | | | |
|---|---|---|---|
| *Dyslipidemia* | | | 0.38 |
| Yes | 706 (88.6%) | 17.8% | |
| No | 91 (11.4%) | 22.6% | |
| *Prior Revascularization Procedure* | | | 0.99 |
| Yes | 358 (44.9%) | 18.3% | |
| No | 439 (55.1%) | 18.4% | |
| *Prior Myocardial Infarction* | | | 0.31 |
| Yes | 176 (22.1%) | 21.4% | |
| No | 621 (77.9%) | 17.5% | |
| *Prior stroke/transient ischemic attack* | | | 0.42 |
| Yes | 93 (11.7%) | 22.2% | |
| No | 704 (88.3%) | 17.8% | |
| *Congestive heart failure* | | | 0.32 |
| Yes | 115 (14.4%) | 22.7% | |
| No | 682 (85.6%) | 17.6% | |
| *Atrial fibrillation* | | | 0.97 |
| Yes | 108 (13.6%) | 18.7% | |
| No | 689 (86.4%) | 18.3% | |
| *Chronic kidney disease* | | | 0.47 |
| Yes | 121 (15.2%) | 21.6% | |
| No | 676 (84.8%) | 17.8% | |
| *PHQ-8 Depression Score* | | | <0.001 |
| 0.0 to <5 | 477 (61.5%) | 7.3% | |
| 5 to <10 | 159 (20.5%) | 22.2% | |
| 10 to <15 | 81 (10.5%) | 41.3% | |
| 15 to 24.0 | 58 (7.5%) | 64.8% | |
| *Baseline stress (PSS-4 $\geq$ 6)* | | | <0.001 |
| Yes | 282 (35.8%) | 34.4% | |
| No | 506 (64.2%) | 9.4% | |

ESSI = ENRICHD social support index, PHQ-8 = 8-point Patient Health Questionnaire depression scale, PAQ = Peripheral Artery Disease Questionnaire.

## Discussion

Observational studies describing the adverse cardiovascular outcomes due to chronic stress exposure have been described since the 1970s and have been replicated in diverse clinical settings and in patients across a spectrum of CVD risk [2,17–20]. Mechanisms that could explain the development and progression of CVD due to chronic stress have also been described and include direct pathophysiological effects [21] and indirect pathways through adverse health behaviors [22]. Chronic stress as a construct is germane to individual patient and societal influences and understanding the impact of these factors in its totality is important. We found that 30% of the patients who experience an AMI and 18% of the patients diagnosed with symptomatic PAD continue to suffer from chronic stress over 1-year of follow-up. This was after the initial event (AMI, worsening symptoms of PAD), which could contribute to stress in the first few days. We developed a risk model in these contemporary cohort of patients with CVD to understand the role of patient and environmental factors towards development of chronic stress. These insights could be invaluable for designing and testing novel stress-reduction strategies in at-risk patient populations and to prioritize preventive policies and programs that could help address systemic sources of distress.

We identified 6 predictors of chronic stress in our CVD cohorts, which were age, sex, economic burden related to medical care, general SES, current smoker, and ESSI score. These predictors have been associated with stress in more heterogeneous populations. For example, financial strain has been associated with chronic stress and adverse outcomes [23,24], and age

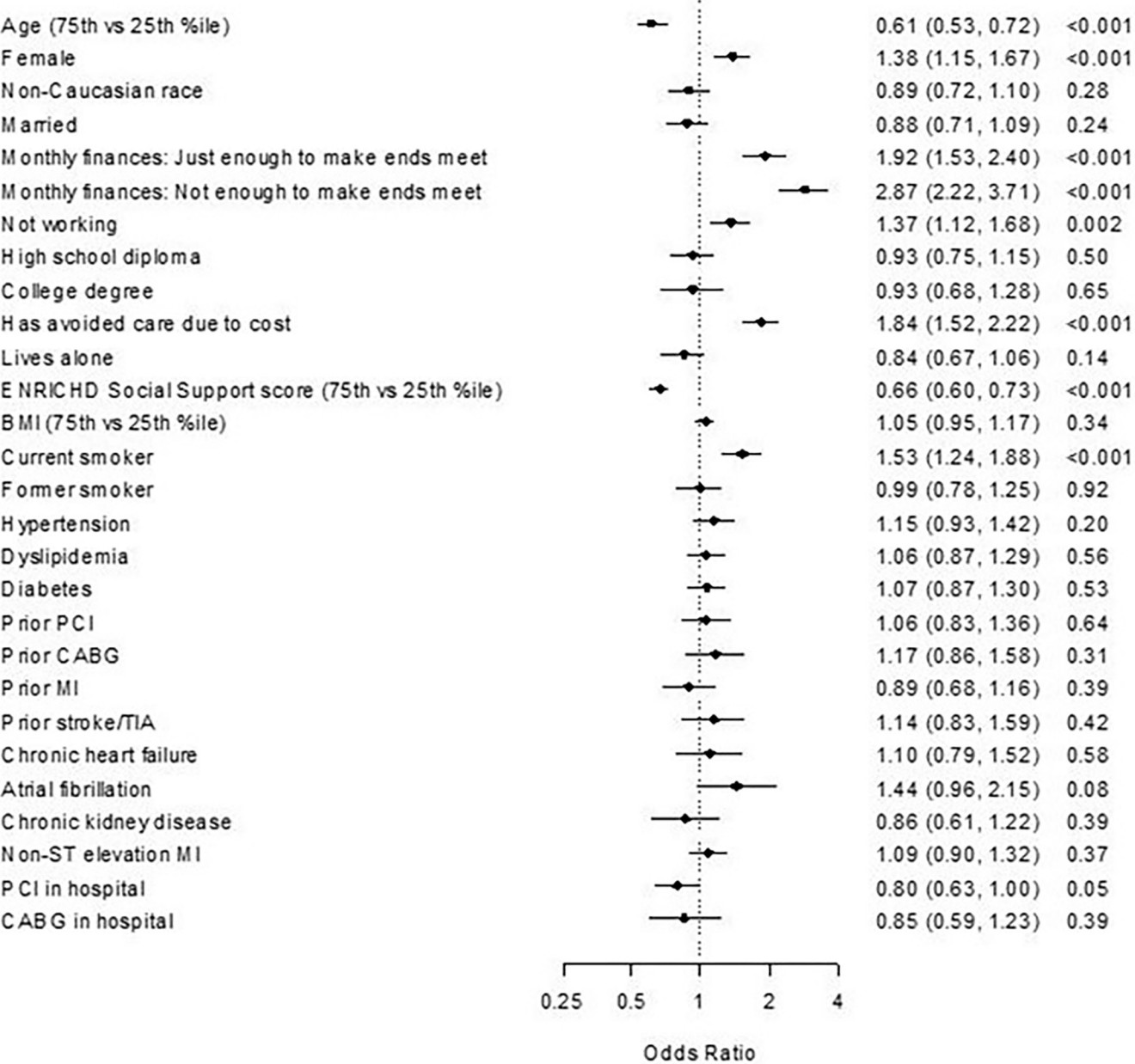

**Fig 1. Independent predictors of chronic stress in the TRIUMPH study.**

has also been shown to influence chronic stress, though the literature here is inconsistent. Some studies have demonstrated that older individuals are less affected by environmental stressors [25,26], while others have demonstrated that they are more vulnerable [27] or have found no association of stress with age [28]. Cognitive theories of aging have postulated that older adults use attentional strategies and reappraisals more frequently to mitigate the impact of environmental influences to avoid chronic stress [29,30]. Moreover, the prevalence of other socioeconomic predictors of chronic stress was higher in younger adults, which could also explain our findings. Smoking has been associated with high stress in several previous studies [31,32]. While the self-medication hypothesis for stress and smoking may explain this link, it is

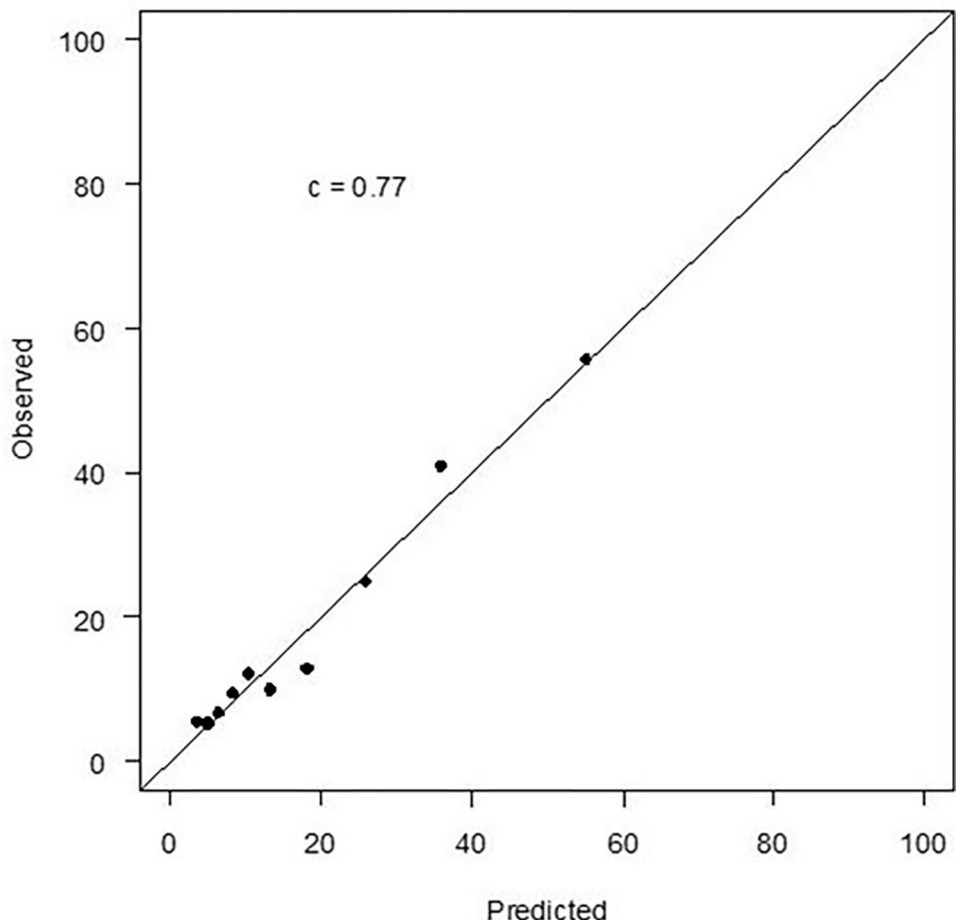

**Fig 2. External validation of the predictive model in the PORTRAIT study.**

also known that smoking is marker of a higher likelihood of chronic stress [33]. This finding was reflected in our results. Furthermore, lack of a social network structure has been linked to anxiety [34] and social support has been shown to buffer the impact of stress on the wellbeing of individuals facing financial hardship [35,36].

From a public health standpoint, it is important to highlight that while public health policies target population health directly through measures such as screening guidelines, immunization etc., social policies to improve SES have also been shown to improve health of the population. For example, in the United Kingdom, implementation of National Minimum Wage legislation in 1999 was associated with improved mental health of low-wage workers [37]. In the US, improvement in SES via programs such as Social Security have had a beneficial impact on the health of the elderly [38]. There is evidence that unconditional universal basic income has positive effect on population health outcomes [39]. In Scotland for example, policies such as citizens basic income were associated with improved population health outcomes [40]. SES, economic burden of health care, and lack of social support were strong predictors of chronic stress in both of our CVD cohorts. Economic plans such as universal basic income, universal coverage of health care, and expansion of social security which would improve SES and the economic burden of medical care for patients with CVD, should be tested to assess its impact as a stress-reduction strategy in patients with CVD.

**Table 3. Socioeconomic predictors of chronic stress in patients stratified by age, in TRIUMPH study.**

| *Socioeconomic Predictors of Chronic Stress* | | | |
|---|---|---|---|
| | **<55 years** **n = 1,633** | **>55 years** **n = 2,707** | **p-value** |
| *Finances at the End of the Month* | | | <0.001 |
| Some money left over | 546 (33.9%) | 1231 (46.5%) | |
| Just enough to make ends meet | 618 (38.3%) | 974 (36.8%) | |
| Not enough to make ends meet | 448 (27.8%) | 441 (16.7%) | |
| Not Working | 540 (33.4%) | 1660 (61.9%) | <0.001 |
| Working | 1078 (66.6%) | 1022 (38.1%) | |
| *Education* | | | <0.001 |
| Less than high school | 313 (19.2%) | 582 (21.6%) | |
| High school | 1047 (64.4%) | 1495 (55.6%) | |
| College degree | 266 (16.4%) | 612 (22.8%) | |
| Has avoided care due to cost | 555 (34.6%) | 555 (34.6%) | <0.001 |
| Has not avoided care due to cost | 1051 (65.4%) | 1051 (65.4%) | |
| *Medical Costs Economic Burden* | | | <0.001 |
| Severe burden | 219 (13.6%) | 228 (8.6%) | |
| Moderate burden | 175 (10.9%) | 234 (8.8%) | |
| Somewhat of a burden | 219 (13.6%) | 288 (10.8%) | |
| A little burden | 136 (8.5%) | 304 (11.4%) | |
| No burden at all | 858 (53.4%) | 1604 (60.3%) | |
| Lives alone | 338 (20.8%) | 723 (27.0%) | <0.001 |
| Does not live alone | 1288 (79.2%) | 1959 (73.0%) | |
| *ESSI score* | | | 0.001 |
| 5 to <20 | 378 (23.8%) | 508 (19.4%) | |
| 20 to <25 | 500 (31.5%) | 828 (31.7%) | |
| >25 | 709 (44.7%) | 1276 (48.9%) | |

Regardless of its trigger or root cause, experiencing chronic stress is linked with an increased cardiovascular risk. It is also known, however, that stress is a modifiable risk factor for which evidence-based management strategies exist [41]. Equipping patients with coping skills to reduce stress in their lives has been shown to be effective in improving quality of life in patients with coronary artery disease [21]. Furthermore, chronic stress management through cognitive behavioral therapy programs [7], as well as through transcendental meditation [8], in addition to standard care, has been shown to reduce the risk of recurrent cardiovascular events in patients with coronary artery disease and to prolong women's life following an acute myocardial infarction. Given the strength of the association found in our study and the fact that this risk factor has been largely ignored in the CVD population, there is an important need for future studies to test the efficacy of stress management strategies on cardiovascular outcomes. The current work can help identify patients who would be most likely to benefit from such interventions.

Our study also had some limitations. Both derivation and validation data sets were obtained from prospective registries that included carefully selected institutions. Whether the enrolling institutes for both TRIUMPH and PORTRAIT studies are representative of other sites not included in these studies is not known. Second, societal, cultural and sociopolitical influences are unique to the US cohorts under study, and whether the major predictors of chronic stress are similar in other countries remains an important area of further work. Third, we quantified stress using the PSS-4 which is a generic and brief instrument to assess perceived stress levels in communities and this measurement may not necessarily extend to other measures of stress or other domains of mental health functioning, nor should it be used for diagnosing purposes, as stress reactions are universal responses. Fourth, the derivative and validation cohorts differed in terms of demographics, socioeconomic conditions and vascular disease (coronary

artery disease vs PAD). However, both disease processes are a manifestation of the same patho-physiological mechanism of atherosclerosis. Moreover, patients with PAD have a similar risk of adverse cardiovascular events (myocardial infarction, stroke), compared to patients diagnosed with coronary artery disease [42]. Our model had good predictive ability in both cohorts, underscoring the value of our model in screening for higher levels of stress in patients across the spectrum of cardiovascular disease. Fifth, it is known that factors at the workplace are associated with risk of experiencing chronic stress and have been associated with development and progression of CVD [1]. Stress at the work-place along with other factors such as economic hardship, lack of social support etc., are important sources of stress for patients with CVD [43,44]. Our aim was to identify patients who are at risk of CVD, and we did not further explore individual sources of stress. Indeed, identifying unique sources of stress in patients with CVD, with the aim of formulating actionable coping strategies, in addition to identifying policy measures that could address some of the more systematic root causes of stress remain areas for future work Finally, approximately 25% of patients had missing follow-up PSS-4 scores due to loss to follow-up, which is similar to missing follow-up rates seen in other prospective AMI registries [45]. We used multiple imputation to account for missing data, but there remains potential for bias. However this still remains an important limitation of our work".

## Conclusion

A substantial number of patients with CVD suffer from chronic stress. We describe and externally validated a well performing prediction model that prognosticates the risk of chronic stress in patients with CVD. As exposure to chronic stress has been linked to adverse clinical outcomes in this population, our model provides valuable insights into the identification of patient-level and societal predictors that could inform design and testing of preventive programs in conjunction with targeted stress-reduction strategies and the patients that may benefit from these in the future.

## Supporting information

**S1 Fig. Calibration plot for the 6-item model in the TRIUMPH study.**
(JPG)

**S1 Table. Model coefficients for all factors in the final model to predict chronic stress in patients with cardiovascular disease.**
(DOCX)

## Author Contributions

**Conceptualization:** Ali O. Malik, Andy Tran, John A. Spertus, Kim G. Smolderen.

**Data curation:** Philip G. Jones, John A. Spertus, Kim G. Smolderen.

**Formal analysis:** Philip G. Jones.

**Funding acquisition:** Philip G. Jones, John A. Spertus, Kim G. Smolderen.

**Investigation:** Vittal Hejjaji, Andy Tran, John A. Spertus.

**Methodology:** Ali O. Malik, Vittal Hejjaji, Andy Tran, John A. Spertus.

**Supervision:** Ali O. Malik.

**Validation:** Ali O. Malik.

**Visualization:** Ali O. Malik.

**Writing – original draft:** Ali O. Malik, Kim G. Smolderen.

**Writing – review & editing:** Ali O. Malik, Carlos Mena-Hurtado, Matthew M. Burg, Mehdi H. Shishehbor, Vittal Hejjaji, Andy Tran, John A. Spertus, Kim G. Smolderen.

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
