## [Decision Letter · Decision Letter 0]

17 Mar 2022

PONE-D-21-13011

Derivation and Validation of a Predictive Model for Chronic Stress in Patients with Cardiovascular Disease

PLOS ONE

Dear Dr. Smolderen,

Thank you for submitting your manuscript to PLOS ONE. After careful consideration, we feel that it has merit but does not fully meet PLOS ONE’s publication criteria as it currently stands. Therefore, we invite you to submit a revised version of the manuscript that addresses the points raised during the review process.

Please revise your manuscript in accordance with the reviewer's comments:

It is of course elegant to establish the prediction model in one cohort and then apply it on another one. However, I feel that the authors have not discussed the significance of the obvious differences between the two cohorts. The PAD cohort is older and among participants only one fourth are working whereas in the TRIUMPH CAD cohort the participants are younger and half them are working. Coronary heart disease is associated with threat to life whereas PAD is associated with pain suffering and severe limitations in movement ability. However for those who are working (in both groups) working conditions are an important potential source of chronic stress. A substantial part of the literature on psychosocial prediction of CAD is based upon stress at work studies. This pertains for instance to the review by Kivimäki and Steptoe that the authors refer to. In fact most of the studies that they refer to deal with stress at work. In addition the international IPD study fairly recently reported from a prospective study that job strain significantly predicts development of PAD (Heikkilä et al 2020). It is odd to see the complete lack of reference to this whole literature.

I can see a point in using the short four-item stress questionnaire for screening purposes, with subsequent more detailed discussion regarding working conditions in individual cases. The authors mention that the questionnaire captures factors associated with both environmental and individual (coping) factors. For less informed readers it may be helpful to explain that follow-up interviews may help in determining a strategy for dealing with the patient´s chronic stress and that things to do might have to do either with coping strategies or family/work organization questions.

The authors hint at the possibility that chronic stress during the follow-up period might have been a consequence of medical procedures and changes in disease course. I think that point needs to be developed because it would be of interest to clinicians reading the paper. Conversely, another topic the authors do not discuss is the potential role of adverse psychosocial conditions in aggravating the clinical course

Despite your professional efforts to explore effects of drop-out on different levels I still think you should discuss more openly the fact that a lot of subjects have fallen behind in several drop-out steps. The ones remaining are likely to be under less chronic stress than others.

We look forward to receiving your revised manuscript.

Kind regards,

M. Harvey Brenner, PhD

Academic Editor

PLOS ONE

Journal Requirements:

2. Thank you for including your ethics statement:  "For both TRIUMPH and PORTRAIT studies, all study participants provided written or telephonic informed consent and the study protocol was approved by Institution Review Boards of Saint Luke’s Hospital and all participating sites."

5. Please note that in order to use the direct billing option the corresponding author must be affiliated with the chosen institute. Please either amend your manuscript to change the affiliation or corresponding author, or email us at plosone@plos.org with a request to remove this option.

6. We noticed you have some minor occurrence of overlapping text with the following previous publication(s), which needs to be addressed:

- https://mospace.umsystem.edu/xmlui/handle/10355/71054

- https://linkinghub.elsevier.com/retrieve/pii/S0022399920308758

In your revision ensure you cite all your sources (including your own works), and quote or rephrase any duplicated text outside the methods section. Further consideration is dependent on these concerns being addressed.

Reviewers' comments:

Reviewer's Responses to Questions

**Comments to the Author**

1. Is the manuscript technically sound, and do the data support the conclusions?

Reviewer #1: Yes

2. Has the statistical analysis been performed appropriately and rigorously? 

Reviewer #1: Yes

3. Have the authors made all data underlying the findings in their manuscript fully available?

Reviewer #1: No

4. Is the manuscript presented in an intelligible fashion and written in standard English?

Reviewer #1: Yes

5. Review Comments to the Author

Reviewer #1: It is of course elegant to establish the prediction model in one cohort and then apply it on another one. However, I feel that the authors have not discussed the significance of the obvious differences between the two cohorts. The PAD cohort is older and among participants only one fourth are working whereas in the TRIUMPH CAD cohort the participants are younger and half them are working. Coronary heart disease is associated with threat to life whereas PAD is associated with pain suffering and severe limitations in movement ability. However for those who are working (in both groups) working conditions are an important potential source of chronic stress. A substantial part of the literature on psychosocial prediction of CAD is based upon stress at work studies. This pertains for instance to the review by Kivimäki and Steptoe that the authors refer to. In fact most of the studies that they refer to deal with stress at work. In addition the international IPD study fairly recently reported from a prospective study that job strain significantly predicts development of PAD (Heikkilä et al 2020). It is odd to see the complete lack of reference to this whole literature.

In view of the great importance of working conditions you might in the future use one questionnaire for working and a slightly different one for a non-working group.

I can see a point in using the short four-item stress questionnaire for screening purposes, with subsequent more detailed discussion regarding working conditions in individual cases. The authors mention that the questionnaire captures factors associated with both environmental and individual (coping) factors. For less informed readers it may be helpful to explain that follow-up interviews may help in determining a strategy for dealing with the patient´s chronic stress and that things to do might have to do either with coping strategies or family/work organization questions.

The authors hint at the possibility that chronic stress during the follow-up period might have been a consequence of medical procedures and changes in disease course. I think that point needs to be developed because it would be of interest to clinicians reading the paper. Conversely, another topic the authors do not discuss is the potential role of adverse psychosocial conditions in aggravating the clinical course

Despite your professional efforts to explore effects of drop-out on different levels I still think you should discuss more openly the fact that a lot of subjects have fallen behind in several drop-out steps. The ones remaining are likely to be under less chronic stress than others.

6. PLOS authors have the option to publish the peer review history of their article (what does this mean?). If published, this will include your full peer review and any attached files.

Reviewer #1: **Yes: **Tores Theorell

---

## [Author Response · Author response to Decision Letter 0]

3 Jun 2022

Please note: Below, we have copied the reviewers’ comments (in bold) and provided our responses in plain text. Sentences added to the manuscript in response to an editor or reviewer comment are indented and placed in italic font. ________________________________________

Reviewer Comments: 

It is of course elegant to establish the prediction model in one cohort and then apply it on another one. However, I feel that the authors have not discussed the significance of the obvious differences between the two cohorts. The PAD cohort is older and among participants only one fourth are working whereas in the TRIUMPH CAD cohort the participants are younger and half them are working. Coronary heart disease is associated with threat to life whereas PAD is associated with pain suffering and severe limitations in movement ability. 

RESPONSE# 1: These are important points. The reviewer is correct in pointing out the differences between the derivative cohort (TRIUMPH study) and validation cohort (PORTRAIT study), with the obvious difference of vascular bed involved, coronary in TRIUMPH cohort and peripheral in PORTRAIT cohort.

From a pathophysiological standpoint, the two conditions coronary artery disease (CAD) and peripheral artery disease (PAD) have the same underlying etiology, i.e., atherosclerosis as a generalized process. Even though the manifestation of symptoms is different, prior work has demonstrated that the clinical event burden in terms of future risk of myocardial infarction and stroke, is similar, if not greater amongst those with PAD vs. CAD (JAMA. 2007 Mar 21;297(11):1197-206). So even though the cohorts represent different arterial beds that got affected, they all represent manifestations of the same underlying disease. It is reassuring to know that the performance of the models across these populations is comparable, which would accommodate screening for high levels of chronic stress throughout atherosclerotic disease trajectories that patients may face. This point is underscored by the c-statistic for our 6-item model which was comparable in both cohorts, 0.75 in TRIUMPH cohort and 0.77 in the PORTRAIT cohort. 

The following sentences have been added to the discussion section. 

Page 12 Line 22-23, Page 13 Line 1-5

“Fourth, the derivative and validation cohorts differed in terms of demographics, socioeconomic conditions and vascular disease (coronary artery disease vs PAD). However, both disease processes are a manifestation of the same pathophysiological mechanism of atherosclerosis. Moreover, patients with PAD have a similar risk of adverse cardiovascular events (myocardial infarction, stroke), compared with patients affected by coronary artery disease.42 Our model had good predictive ability in both cohorts, underscoring the value of our model in screening for higher levels of stress in patients across the spectrum of cardiovascular disease. “

However for those who are working (in both groups) working conditions are an important potential source of chronic stress. A substantial part of the literature on psychosocial prediction of CAD is based upon stress at work studies. This pertains for instance to the review by Kivimäki and Steptoe that the authors refer to. In fact most of the studies that they refer to deal with stress at work. In addition the international IPD study fairly recently reported from a prospective study that job strain significantly predicts development of PAD (Heikkilä et al 2020). It is odd to see the complete lack of reference to this whole literature. In view of the great importance of working conditions you might in the future use one questionnaire for working and a slightly different one for a non-working group.

RESPONSE# 2: We agree with the reviewer. Certainly, some of the literature exploring the impact of stress on development and progression of cardiovascular disease pertains to stress at the workplace. We have acknowledged this in our revision and referred to the work by Heikkilä et al. 

It is indeed important to understand sources of stress for patients with cardiovascular disease. Stress at workplace is indeed an important consideration. However other sources of stress such as economic hardship, lack of social support also contribute to the overall burden of stress in these patients (J Am Coll Cardiol. 2018 Jun 5;71(22):2585-2597, Heart. 2020 Sep;106(18):1394-1399). Future work should focus on identifying these causes of stress to help identify interventions to help patients cope, as a strategy to improve outcomes, in addition to identifying policy measures that could address some of the more systematic root causes of stress. While we did not explore source of stress with aim of formulating actionable coping mechanisms, this remains an area for future work. 

The following sentences have been added to the discussion section. 

Page 13 Line 5-13

“Fifth, it is known that factors at the workplace are associated with risk of experiencing chronic stress and have been associated with development and progression of CVD.1 Stress at the work-place along with other factors such as economic hardship, lack of social support etc., are important sources of stress for patients with CVD.43,44 Our aim was to identify patients who are at risk of CVD, and we did not further explore individual sources of stress. Indeed, identifying unique sources of stress in patients with CVD, with the aim of formulating actionable coping strategies, in addition to identifying policy measures that could address some of the more systematic root causes of stress remain areas for future work”. 

I can see a point in using the short four-item stress questionnaire for screening purposes, with subsequent more detailed discussion regarding working conditions in individual cases. The authors mention that the questionnaire captures factors associated with both environmental and individual (coping) factors. For less informed readers it may be helpful to explain that follow-up interviews may help in determining a strategy for dealing with the patient´s chronic stress and that things to do might have to do either with coping strategies or family/work organization questions.

RESPONSE# 3: This is a very important point. As next steps, we will need to further identify the sources of stress patients affected by cardiovascular disease are dealing with (e. g. work stress, loneliness, economic hardship, as well as stress from navigating the disease and its management itself etc.) and develop tailored interventions to manage or mitigate sources of stress. This remains an area of future work. Please see RESPONSE# 2. 

The authors hint at the possibility that chronic stress during the follow-up period might have been a consequence of medical procedures and changes in disease course. I think that point needs to be developed because it would be of interest to clinicians reading the paper. Conversely, another topic the authors do not discuss is the potential role of adverse psychosocial conditions in aggravating the clinical course.

RESPONSE# 4: The reviewer is correct. We were cognizant of this issue and agree that stress during follow-up could be due to medical procedures and changes in disease course. To account for this bias, we excluded baseline stress assessment from the definition of chronic stress. We had mentioned that in the methods section and have clarified this further in the revised version of the manuscript. For TRIUMPH study (derivation cohort), follow-up assessments were at 1,6- and 12-month follow-up intervals. For the PORTRAIT study (validation cohort) follow-up assessments were at 3,6 and 12-month follow-up intervals. We defined chronic stress as two or more follow-up assessments with PSS-4 score ≥ 6. 

The following sentences were added/modified to the manuscript. 

Page 6 Line 2-7

“We wanted to quantify a patient’s exposure to chronic stress, during the 12-months of follow-up. Therefore, to provide more stable categorization of stress levels at follow-up we defined chronic stress as 2 or more follow-up PSS-4 assessments of �6, after the initial baseline assessment. As the initial event (AMI or worsening PAD symptoms) could contribute to the patient’s stress in the first few days, baseline PSS-4 assessments were not included in the definition.” 

Page 10 Line 10-11

“This was after the initial event (AMI, worsening symptoms of PAD), which could contribute to stress in the first few days.”

Despite your professional efforts to explore effects of drop-out on different levels I still think you should discuss more openly the fact that a lot of subjects have fallen behind in several drop-out steps. The ones remaining are likely to be under less chronic stress than others.

RESPONSE# 5: Thank you for the suggestion. We agree that missing data is a big limitation. Even though we used multiple imputations by chained equations (MICE) [Journal of statistical software 2010: 1-68], with predictive mean matching to account for any bias, this remains an important limitation of our work. We have highlighted this further in the limitations section of our revised manuscript. 

The following sentences have been added to the manuscript. 

Page 13 Line 15-16

“We used multiple imputation to account for missing data, but there remains potential for bias. However, this still remains an important limitation of our work”

---

## [Decision Letter · Decision Letter 1]

23 Sep 2022

Derivation and Validation of a Predictive Model for Chronic Stress in Patients with Cardiovascular Disease

PONE-D-21-13011R1

Dear Dr. Smolderen 

We’re pleased to inform you that your manuscript has been judged scientifically suitable for publication and will be formally accepted for publication once it meets all outstanding technical requirements.

Kind regards,

Xianwu Cheng, M.D., Ph.D., FAHA

Academic Editor

PLOS ONE

Additional Editor Comments (optional):

Authors have adressed all original concerns.

Reviewers' comments:

Reviewer's Responses to Questions

**Comments to the Author**

1. If the authors have adequately addressed your comments raised in a previous round of review and you feel that this manuscript is now acceptable for publication, you may indicate that here to bypass the “Comments to the Author” section, enter your conflict of interest statement in the “Confidential to Editor” section, and submit your "Accept" recommendation.

Reviewer #1: All comments have been addressed

2. Is the manuscript technically sound, and do the data support the conclusions?

Reviewer #1: Yes

3. Has the statistical analysis been performed appropriately and rigorously? 

Reviewer #1: Yes

4. Have the authors made all data underlying the findings in their manuscript fully available?

Reviewer #1: Yes

5. Is the manuscript presented in an intelligible fashion and written in standard English?

Reviewer #1: Yes

6. Review Comments to the Author

Reviewer #1: The authors have been honest about weaknesses and discuss them adequately. They do contribute to the literature

7. PLOS authors have the option to publish the peer review history of their article (what does this mean?). If published, this will include your full peer review and any attached files.

Reviewer #1: **Yes: **Tores Theorell

---

## [Editor Report · Acceptance letter]

7 Oct 2022

PONE-D-21-13011R1 

 Derivation and Validation of a Predictive Model for Chronic Stress in Patients with Cardiovascular Disease 

Dear Dr. Smolderen:

I'm pleased to inform you that your manuscript has been deemed suitable for publication in PLOS ONE. Congratulations! Your manuscript is now with our production department. 

Kind regards, 

on behalf of

Associate Prof. Xianwu Cheng 

Academic Editor

PLOS ONE